# A systematic review of federated learning applications for biomedical data

Matthew G. Crowson[1,2]*, Dana Moukheiber[3], Aldo Robles Arévalo[4,5], Barbara D. Lam[6], Sreekar Mantena[7], Aakanksha Rana[8], Deborah Goss[1], David W. Bates[9,10], Leo Anthony Celi[11,12]

1 Department of Otolaryngology-Head & Neck Surgery, Massachusetts Eye & Ear, Boston, Massachusetts, United States of America, 2 Department of Otolaryngology-Head & Neck Surgery, Harvard Medical School, Massachusetts, United States of America, 3 Laboratory for Computational Physiology, Massachusetts Institute of Technology, Cambridge, MA, United States of America, 4 IDMEC, Instituto Superior Técnico, Universidade de Lisboa, Lisbon, Portugal, 5 Data & Analytics, NTT DATA Portugal, Lisbon, Portugal, 6 Department of Hematology & Oncology, Beth Israel Deaconess Medical Center, Boston, Massachusetts, United States of America, 7 Harvard College, Boston, Massachusetts, United States of America, 8 Massachusetts Institute of Technology, Boston, Massachusetts, United States of America, 9 Division of General Internal Medicine and Primary Care, Brigham and Women's Hospital, Boston, MA, United States of America, 10 Department of Health Policy and Management, Harvard T. H. Chan School of Public Health, Boston, MA, United States of America, 11 Institute for Medical Engineering and Science, Massachusetts Institute of Technology, Cambridge, Massachusetts, United States of America, 12 Division of Pulmonary, Critical Care and Sleep Medicine, Beth Israel Deaconess Medical Center, Boston, Massachusetts, United States of America

* matthew_crowson@meei.harvard.edu

**Data Availability Statement:** The data for this systematic review is derived from a qualitative assessment of publicly available published papers/

## Abstract

### Objectives

Federated learning (FL) allows multiple institutions to collaboratively develop a machine learning algorithm without sharing their data. Organizations instead share model parameters only, allowing them to benefit from a model built with a larger dataset while maintaining the privacy of their own data. We conducted a systematic review to evaluate the current state of FL in healthcare and discuss the limitations and promise of this technology.

### Methods

We conducted a literature search using PRISMA guidelines. At least two reviewers assessed each study for eligibility and extracted a predetermined set of data. The quality of each study was determined using the TRIPOD guideline and PROBAST tool.

### Results

13 studies were included in the full systematic review. Most were in the field of oncology (6 of 13; 46.1%), followed by radiology (5 of 13; 38.5%). The majority evaluated imaging results, performed a binary classification prediction task via offline learning (n = 12; 92.3%), and used a centralized topology, aggregation server workflow (n = 10; 76.9%). Most studies were compliant with the major reporting requirements of the TRIPOD guidelines. In all, 6 of 13 (46.2%) of studies were judged at high risk of bias using the PROBAST tool and only 5 studies used publicly available data.

manuscripts. The manuscripts referenced in this paper are in the public domain.

**Funding:** Dr. Bates reports grants and personal fees from EarlySense, personal fees from CDI Negev, equity from ValeraHealth, equity from Clew, equity from MDClone, personal fees and equity from AESOP, personal fees and equity from FeelBetter, and grants from IBM Watson Health, outside the submitted work. The funders had no role in study design, data collection and analysis, decision to publish, or preparation of the manuscript

**Competing interests:** I have read the journal's policy and the authors of this manuscript have the following competing interests: Leo Anthony Celi is the Editor-in Chief of PLOS Digital Health.

## Conclusion

Federated learning is a growing field in machine learning with many promising uses in healthcare. Few studies have been published to date. Our evaluation found that investigators can do more to address the risk of bias and increase transparency by adding steps for data homogeneity or sharing required metadata and code.

### Author summary

Interest in machine learning as applied to challenges in medicine has seen an exponential rise over the past decade. A key issue in developing machine learning models is the availability of sufficient high-quality data. Another related issue is a requirement to validate a locally trained model on data from external sources. However, sharing sensitive biomedical and clinical data across different hospitals and research teams can be challenging due to concerns with data privacy and data stewardship. These issues have led to innovative new approaches for collaboratively training machine learning models without sharing raw data. One such method, termed 'federated learning,' enables investigators from different institutions to combine efforts by training a model locally on their own data, and sharing the parameters of the model with others to generate a central model. Here, we systematically review reports of successful deployments of federated learning applied to research problems involving biomedical data. We found that federated learning links research teams around the world and has been applied to modelling in such as oncology and radiology. Based on the trends we observed in the studies reviewed in our paper, we observe there are opportunities to expand and improve this innovative approach so global teams can continue to produce and validate high quality machine learning models.

## Introduction

Machine learning (ML) requires high quality datasets to produce unbiased and generalizable models. While there have been collaborative initiatives to create large data repositories (e.g. Observational Health Data Sciences and Informatics, IBM Merge Healthcare, Health Data Research UK), these are challenging to implement and maintain because of technical and regulatory barriers.[1] Another key challenge for the development of robust ML models is the requirement to validate the model on data from external sources. However, sharing sensitive biomedical and clinical data across separate institutions can be challenging due to concerns with data privacy and stewardship. Federated learning (FL) offers a promising solution to these challenges, particularly in healthcare where patient data privacy is paramount.

First developed in the mobile telecommunications industry, FL allows multiple separate institutions to collaboratively develop a ML algorithm by sharing the model and its parameters rather than the training data.[2] In this development paradigm, institutions maintain control over their data while realizing the benefit of a model that has been trained and validated using diverse data across multiple institutions. This collaborative approach is important not only for increasing the scope of academic research partnerships, but also for the development and implementation of robust ML models trained on disparate data. In addition to producing robust ML models, FL may enable more equitable precision medicine. Combining data from regional, national, or international institutions could benefit patients from underrepresented

groups, patients with orphan diseases, and hospitals with fewer resources by providing access to point-of-care ML algorithms.

The potential for FL to accelerate robust ML model development and precision medicine has led to an increasing volume of scholarly reports on FL system proof-of-concept and validation in the past several years. The power of these collaborative models was demonstrated during the COVID-19 pandemic, when multiple groups used FL models to improve quality of care and outcomes. [3] Larger initiatives to bring FL to the bedside, such as the Federated Tumor Segmentation Initiative, are also underway. [4] Despite the tremendous potential of FL, there are still concerns around data quality and standardization as well as barriers to adoption. [5]

In this systematic review, our objective was to evaluate the current state of FL in medicine by evaluating ML algorithms that were developed and validated using a FL framework. We explored and compared the types of FL architectures deployed, clinical applicability and value, predictive performance, and the quality of the scholarly reports in terms of best practices for ML model development. We also discuss the overall strengths and limitations of FL in medicine at present with a forecast on opportunities and barriers for the future of FL.

## Methods

The review was conducted in accordance with the Cochrane Handbook for Systematic Reviews of Interventions and reporting requirements outlined by Preferred Reporting Items for Systematic Reviews and Meta-Analyses (**S1 Table**). [6]

### Search strategy & selection criteria

In this systematic review, we searched for published studies that developed or validated a FL framework for predictive modeling of diagnoses, treatment outcomes or prognostication for any disease entity using biomedical data. Methods of analysis and inclusion criteria were specified in advance. The systematic review of the literature used a controlled vocabulary and keyword terms relating to the collaborative use of artificial intelligence in medicine such as "machine learning," "federated learning," "distributed learning," "electronic medical record," "health data," and "data exchange" (**S2 Table**). We searched Ovid MEDLINE (1946-), Embase. com (1947-), Web of Science Core Collection (1900-), CINAHL (1937-), and ACM Digital Library (1908-). The PRISMA guidelines were used to document the search.[6] All of the searches were designed and conducted by a reference librarian (DG). The search was reviewed by a second librarian. No language or date limitations were used. The final searches were run on October 29, 2020.

Each study eligibility was assessed by at least two reviewers (two of M.C., L.C., B.L., A.A., S. M., A.R., D.M.) who independently screened titles and abstracts of the search results. Non-consensus cases were resolved by a third reviewer. We excluded studies that used simulated distributed learning (not 'actual' geographically separate nodes), data that were not biomedical in nature, non-English language writing, review-style or editorial papers, papers with no full-text available, and papers that did not report clinical outcomes or applicability.

### Data extraction

Data extraction was completed using a predefined data extraction tool. The tool addressed several domains including study design (number of participating sites/nodes, countries, medical/biomedical subspeciality, ML model prediction task), study data (data type, dataset size, number of features/variables, missing data imputation approach), ML modeling approach (ML algorithm used, model justification, performance metrics, hyperparameter tuning, model calibration, model validation strategy, model interpretability, potential for ML model bias), FL

system approach (FL architecture chosen, FL topology, computing plan for nodes), and research reproducibility (source code availability, container availability).

## Quality assessment

To assess the quality of the reporting of the ML approach in each included study, we used the Transparent Reporting of a multivariable prediction model for Individual Prognosis or Diagnosis (TRIPOD) guideline. [7] The TRIPOD guideline was developed as a consensus framework to appraise the reporting of studies developing or validating a diagnostic or prognostic prediction model. To assess the risk-of-bias of the included studies, we utilized the Prediction model Risk Of Bias ASsessment Tool (PROBAST).[8] PROBAST was developed as a tool for systematic reviews to assess the risk-of-bias and applicability of studies describing diagnostic and prognostic prediction models. We chose PROBAST over other commonly used systematic review risk-of-bias tools as the scope of our review is limited to predictive (i.e., machine learning) models.

## Results

Our search strategy identified 2173 records and 2171 were screened after removal of duplicates (**Fig 1**). Of the screened articles, 99 full-text articles were assessed for eligibility based on our criteria. Thirteen studies were included in the full systematic review.

## General study characteristics

All included studies involved at least two participating institutions (i.e., 'nodes'), with the largest collaborative effort comprising data from 50 different hospitals/institutions [9] (**Table 1**).

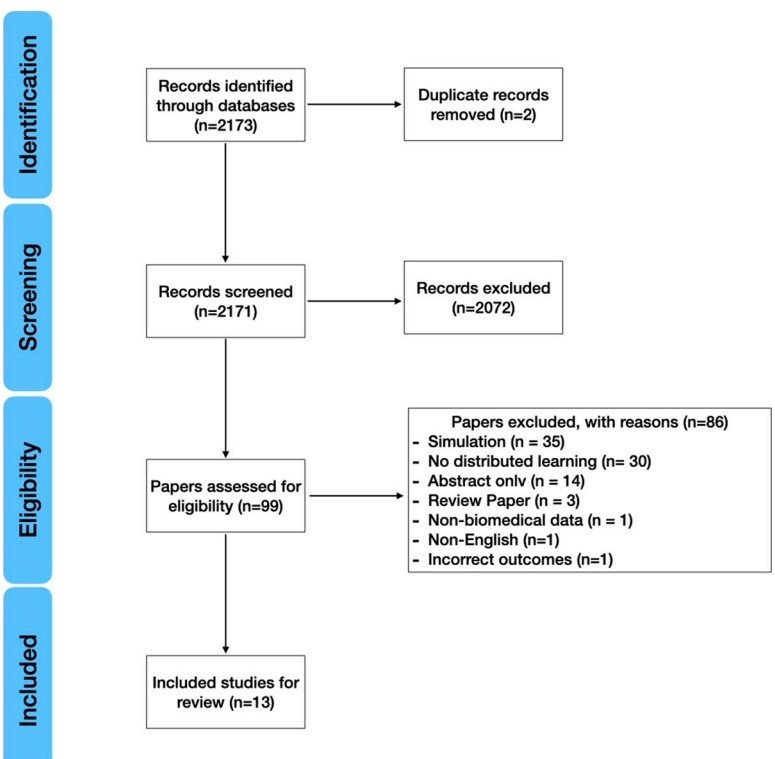

**Fig 1. Study inclusion and exclusion flow.**

**Table 1. Basic characteristics of included studies.**

| Citation | Interdisciplinary Collaboration | Participating Sites/Institutions | Participating Countries | Medical or Biomedical Subspecialty | Predicted Outcome/ Target | Detailed Patient Inclusion/Exclusion Criteria |
|---|---|---|---|---|---|---|
| Bogowicz et al., (2020) [10] | Yes | 5 | (4) Canada, Netherlands, Switzerland, United States | (4) Radiology, Radiation Oncology, Medical Oncology, Otolaryngology | (1) Survival | Yes |
| Deist et al., (2020) [11] | No | 8 | (5) China, England, Italy, Netherlands, Wales | (2) Medical Oncology, Radiation Oncology | (1) Survival | Yes |
| Deist et al., (2017) [12] | No | 5 | (3) Belgium, Germany, Netherlands | (1) Radiation Oncology | (1) Post-radiotherapy dyspnea grade | Yes |
| Huang et al., (2019) [9] | Yes | 50 | (1) United States | (1) Critical Care | (2) In-Hospital Mortality, ICU Stay Time | No |
| Jochems et al, (2017) [13] | Yes | 3 | (3) Netherlands, United States, United Kingdom | (1) Radiation Oncology | (1) Survival | No |
| Jochems et al. (2016) [14] | Yes | 5 | (3) Belgium, Germany, Netherlands | (1) Radiation Oncology | (1) Dyspnea | Yes |
| Li J et al., (2020) [15] | Yes | 16 | (1) United States, China | (1) Medical Oncology | (1) Survival | Yes |
| Li X et al., (2020) [16] | Yes | 4 | (1) United States | (2) Radiology, Psychiatry, Radiology | (1) Autism Spectrum Disorder Classification | No |
| Remedios et al., (2019) [17] | Yes | 2 | (1) United States | (1) Radiology | (1) Hemorrhage or Hematoma Segmentation | Yes |
| Remedios et al., (2020) [18] | Yes | 5 | (1) United States | (1) Radiology | (1) Hemorrhage or Hematoma Segmentation | Yes |
| Sheller et al., (2020) [19] | Yes | 12 | (1) United States | (2) Medical Oncology, Radiology | (1) Tumor Detection | Yes |
| Tian, et al., (2020) [20] | Yes | 6 | (2) China, United States | (2) Gastroenterology, Medical Oncology | (1) Survival | Yes |
| Xu, et al., (2020) [21] | Yes | 4 | (1) China | (3) Infectious Disease, Radiology, Pulmonology | (1) COVID-19 Diagnosis | No |

All the studies were performed with interdisciplinary teams composed of clinicians and technical experts (i.e., data scientists, data engineers). All studies were completed in developed countries, with most studies completed in an international collaborative setting (n = 7, 53.9%), followed by studies performed exclusively in the United States (n = 5, 38.5%). The most common clinical subspeciality represented was medical and radiation oncology (n = 6, 46.1%) followed by radiology (n = 5, 38.5%). Cancer prognostication was the most common use case (n = 5, 38.5%), followed by pathology identification using imaging (n = 4, 30.8%).

## Dataset characteristics

Dataset sizes of study subjects or derived data ranged from hundreds to tens of thousands (**Table 2**). The majority (n = 8; 61.5%) utilized structured data. When reported, the models comprised feature counts ranging from 5 to 1,400. Only 3 (23.1%) studies included a detailed description of the inclusion criteria of study subjects. Publicly available data sources were used in 2 (15.4%) studies. Most studies cited use of a non-public data source, and only a small number of teams made their data publicly available as part of their initial manuscript submission (n = 3, 23.1%). Missing data and imputation methods were reported in the models that utilize

**Table 2. Data composition and characteristics of included studies.**

| Citation | Data Type | Patient Cohort Size | Data Publicly Available | Number of Model Features | Modeling Dataset Size | Cohort Descriptive Statistical Analysis | Missing Data Management |
|---|---|---|---|---|---|---|---|
| Bogowicz et al., (2020) [10] | Non-Structured | 1,064 | No | 981 | *Not reported* | Yes | Yes |
| Deist et al., (2020) [11] | Structured | 23,203 | No | 6 | 23,203 patients | Yes | Yes |
| Deist et al., (2017) [12] | Structured | 268 | No | 3 | *Not reported* | Yes | Yes |
| Huang et al., (2019) [9] | Structured | 28,000 | Yes | 1,400 | 28,000 patients | Yes | Yes |
| Jochems et al, (2017) [13] | Structured | 894 | Yes | 9 | *Not reported* | Yes | Yes |
| Jochems et al. (2016) [14] | Structured | 287 | No | 5 | 287 patients | Yes | Yes |
| Li J et al., (2020) [15] | Structured | *Not reported* | Yes; partial | 12 | *Not reported* | *Not reported* | *Not reported* |
| Li X et al., (2020) [16] | Structured | 370 | Yes; partial | *Not reported* | *Not reported* | Yes | *Not reported* |
| Remedios et al., (2019) [17] | Non-Structured | 27 | No | *NA* | 27 CT scans | *Not reported* | Yes |
| Remedios et al., (2020) [18] | Non-Structured | *Not reported* | No | *NA* | 161 CT scans | *Not reported* | Yes |
| Sheller et al., (2020) [19] | Non-structured | 406 | Yes; partial | *NA* | 406 MRI scans | *Not reported* | Yes |
| Tian, et al., (2020) [20] | Structured | 70,906 | No | 12 | 70,906 patients | Yes | Yes |
| Xu, et al., (2020) [21] | Non-Structured | 1,276 | No | *NA* | 5,732 CT images | Yes | *Not reported* |

CT: computed tomography

MRI: magnetic resonance imaging

structured data. For studies using computer vision techniques, image preprocessing techniques were routinely detailed.

## Modeling

Most studies performed a binary classification (n = 11; 84.6%) prediction task via offline learning (n = 12; 92.3%) (**Table 3**). Various model architectures were used spanning basic logistic regression, Bayesian networks, tree-based methods, and deep learning. All studies reported on the performance of their models with most studies reporting the area under the receiver operating characteristic (AUC-ROC) curve for a binary classification task (n = 8; 61.5%). There was considerable heterogeneity in the studies reporting on hyperparameter optimization, validation strategies, and comparison between model architectures when multiple model types were developed. Only one study explicitly explored the potential for bias in their data and modeling workflow. [11]

## Federated learning architecture

We categorized the studies' workflow, topology, and node computing plan using an established FL vocabulary. [1] Most studies used a centralized topology (n = 10; 76.9%), aggregation server workflow (n = 10; 76.9%), and an aggregation server for their computing plan at the nodes

(n = 10; 76.9%) (**Table 4**). Only 7 (53.9%) studies provided access to their models or modeling code (e.g. via Github), and only 1 (7.7%) study provided a model Docker container. [19]

### Quality assessment

Most studies were compliant with the main reporting requirements of the TRIPOD guidelines, [7] except for reporting on methods for handling missing data and reporting on the unadjusted associations between candidate features and the outcome variable (**S3 Table**; **S1 Fig**). The risk-of-bias was assessed using the Prediction model Risk Of Bias ASsessment Tool (PROBAST). Overall, 6 (46.2%) of the studies were judged as having high risk of bias and 6 (46.2%) were judged of high concern for applicability integrating the four PROBAST domains (**S4 Table**; **S2 Fig**).

## Discussion

The potential for federated learning to accelerate machine learning model development and validation has led to great interest in this area and a growing volume of published works reporting proof-of-concept and early implementations. Prior narrative and systematic reviews on FL as applied to healthcare have elaborated on the technical nuances of FL architectures, models, and datasets as well as higher order issues such as legal contexts, privacy, and ethical considerations. [Zerka 2020; Pfitzner 2021; Shyu 2021]. In our systematic review, we add to this existing knowledge by evaluating the current state of FL in biomedical contexts through a search of studies reporting on ML algorithms developed and validated using a FL framework specifically for biomedical data. Several major themes emerged. First, computer vision applications were the predominant use case. Second, most were international collaborations exclusively in developed countries. Third, there was overall a lack of discussion or consideration for actual or potential bias in the study data. Fourth, only approximately half the studies included or referenced code and/or a tool for externally validating their results. Fifth, only one study reported the use of an interoperability framework with respect to data curation. [15] Nonetheless, this approach has great potential, as it allows development of models at multiple sites and protects privacy.

### Computer vision

Early adoption of FL approaches has been led by Radiology, Radiation Oncology, and Medical Oncology (n = 7, 53.8%). These clinical specialties share a common data medium in the form of medical imagery (e.g., medical imaging, pathology slides) which are readily analyzed through computer vision techniques. [22] The propensity for use of computer vision in FL might be influenced by the ease and standardization of image data pre-processing techniques that can be uniformly deployed across participating nodes. Data pre-processing techniques applied to images (e.g., resizing, rescaling, flipping, normalization) before modeling do not require exploration of the whole image data lake, though different modalities or medical device brands still require thoughtful protocol design. For example, bias in image data might arise with the use of different capturing devices.

### Structured vs. unstructured data

Structured data is generally defined as organized or searchable data consisting of numbers and values. Unstructured data types exist in either a native format or no pre-defined format such is the case with images, video, or audio. Studies comprising structured data were relatively limited in this review. Structured data in a biomedical context, such as the fields found in

**Table 3. Modeling and analysis characteristics of included studies.**

| Citation | Modeling Approach | Online or Offline Learning | Algorithm | Algorithm Selection Justification | Algorithm Performance Metric(s) | Hyperparameter Methods Reported | Model Calibration Reported | Validation Methods | Model Performance Comparison | Model Interpretability Reported | Clinical Applicability Reported | Bias Discussed |
|---|---|---|---|---|---|---|---|---|---|---|---|---|
| Bogowicz et al., (2020) [10] | (1) Binary Classification | Offline | (1) Logistic Regression | Not reported | (1) AUC-ROC | Not reported | Yes | Yes | Yes | Yes | Yes | Not reported |
| Deist et al., (2020) [11] | (1) Binary Classification | Offline | (1) Logistic Regression | Interpretability | (2) RMSE, AUC-ROC | Not reported | Yes | Yes | Yes | Yes | Yes | Yes |
| Timo M. Deist et al., (2017) [12] | (1) Binary Classification | Offline | (1) Support Vector Machine | Not reported | (1) AUC-ROC | Yes | Not reported | Yes | Yes | Yes | Yes | Not reported |
| Huang et al., (2019) [9] | (2) Clustering, Binary Classification | Offline | (2) K-Means, Neural Network | Benchmarking to previous studies | (2) AUC-ROC, AUPRC | Yes | Not reported | Yes | Yes | Not reported | Yes | Not reported |
| Jochems et al., (2017) [13] | (1) Multi Classification | Offline | (1) Bayesian Network | Benchmarking to Previous Studies | (1) AUC-ROC | Not reported | Yes | Yes | Yes | Yes | Yes | Not reported |
| Jochems et al. (2016) [14] | (1) Binary Classification | Offline | (1) Bayesian Network | Handling Missing Data | (1) AUC-ROC | Yes | Not reported | Yes | Yes | Yes | Yes | Not reported |
| Li J et al., (2020) [15] | (1) Binary Classification | Offline | (1) Random Forest | Benchmarking to Previous Studies | (1) AUC-ROC | Yes | Yes | Yes | Yes | Yes | Yes | Not reported |
| Li X et al., (2020) [16] | (1) Binary Classification | Offline | (1) Neural Network | Not reported | (1) Class Accuracy | Not reported | Not reported | Yes | Yes | Yes | Yes | Not reported |
| Remedios et al., (2019) [17] | (1) Binary Classification | Offline | (1) Neural Network | Benchmarking to Previous Studies | (1) DSC | Yes | Not reported | Yes | Yes | Not reported | Yes | Not reported |
| Remedios et al., (2020) [18] | (1) Binary Classification | Offline | (1) Neural Network | Benchmarking to Previous Studies | (1) DSC | Yes | Not reported | Yes | Yes | Not reported | Yes | Not reported |
| Sheller et al., (2020) [19] | (1) Binary Classification | Offline | (1) Neural Network | Benchmarking to Previous Studies | (1) DSC | Yes | Not reported | Yes | Yes | Not reported | Yes | Not reported |
| Tian, et al., (2020) [20] | (1) Binary Classification | Online | (1) Logistic Regression | Interpretability | (1) AUC-ROC | Not reported | Yes | Yes | Yes | Yes | Yes/partial | Not reported |
| Xu, et al., (2020) [21] | (1) Multi Classification | Offline | (1) Neural Network | Not reported | (1) AUC-ROC | Yes | Not reported | Yes | Yes | Not reported | Yes | Not reported |

AUC-ROC: Area under the receiving operating characteristic curve

AUPRC: area under the precision-recall curve

DSC: Dice Similarity Coefficient

RMSE: Root mean squared error

**Table 4. Federated learning approach, topologies, and reproducibility of included studies.** Workflow, topologies and computing plan classification adapted from Rieke et al. [1].

| Citation | Federated Learning Workflow[1] | Federated Learning Topology[1] | Computing Plan for Nodes[1] | Code Provided (e.g., GitHub repository) | Containers or Interface/Platform Available for External Validation (e.g., Docker container) |
|---|---|---|---|---|---|
| Bogowicz et al., (2020) [10] | (1) Aggregation Server | (1) Centralized | (1) Aggregation Server | *Not Reported* | *Not Reported* |
| Deist et al., (2020) [11] | (1) Aggregation Server | (1) Centralized | (1) Aggregation Server | Yes | *Not Reported* |
| Timo M. Deist et al., (2017) [12] | (1) Aggregation Server | (1) Centralized | (1) Aggregation Server | *Not Reported* | *Not Reported* |
| Huang et al., (2019) [9] | (1) Aggregation Server | (1) Centralized | (1) Aggregation Server | *Not Reported* | *Not Reported* |
| Jochems et al, (2017) [13] | (1) Aggregation Server | (1) Centralized | (1) Aggregation Server | Yes | *Not Reported* |
| Jochems et al. (2016) [14] | (1) Aggregation Server | (1) Centralized | (1) Aggregation Server | *Not Reported* | *Not Reported* |
| Li J et al., (2020) [15] | (1) Aggregation Server | (1) Centralized | (1) Aggregation Server | *Not Reported* | *Not Reported* |
| Li X et al., (2020) [16] | (1) Aggregation Server | (1) Centralized | (1) Aggregation Server | Yes | *Not Reported* |
| Remedios et al., (2019) [17] | (1) Peer-to-Peer (Cyclical Weight Transfer) | (1) Decentralized (Cyclical Weight Transfer) | (1) Sequential (Cyclical Weight Transfer) | Yes | *Not Reported* |
| Remedios et al., (2020) [18] | (1) Peer-to-Peer (Cyclical Weight Transfer) | (1) Decentralized (Cyclical Weight Transfer) | (1) Sequential (Cyclical Weight Transfer) | Yes | *Not Reported* |
| Sheller et al., (2020) [19] | (2) Peer-to-peer, Aggregation server | (2) Decentralized, Centralized | (1) Sequential (Cyclical Weight Transfer), Aggregation server | Yes | Yes |
| Tian, et al., (2020) [20] | (1) Aggregation Server | (1) Centralized | (1) Aggregation Server | *Not Reported* | *Not Reported* |
| Xu, et al., (2020) [21] | (1) Aggregation Server | (1) Centralized | (1) Aggregation Server | Yes | *Not Reported* |

electronic health records (EHRs), require exploratory data analysis and careful coordination between the participating institutions prior to beginning model training. The crux of this issue is that different organizations may capture information in different ways or may have varying definitions for the same term. For instance, in critical care medicine there exist different validation methods for categorizing and defining 'sepsis.' One hospital may use qSOFA to define sepsis while another uses SIRS criteria. [23] This is a well-recognized challenge in healthcare information technology, and efforts are underway to standardize how organizations capture information so that we can better collaborate on a national and international scale. [24] Recent governmental approaches such as the European Commission aims to create a 'European Data Space' (recent updates available at https://ec.europa.eu/health/ehealth/dataspace_en) are attempting to promote a better exchange and access to different types of health data (EHRs, genomics data, etc.) for Europe-wide healthcare delivery, research, and health policy development. This represents an enormous effort that requires technical and semantic interoperability

between the different infrastructures and IT systems among their member states. Even with standardized fields and definitions, researchers will still need to contend with the accuracy of the data captured. Discrete fields may encourage more standardized responses, but oftentimes the richest EHR data is found in free-text fields that are prone to error and addenda.

## Bias–potential or actual

Success of FL is predicated on assumptions of consistent data curation across the participating nodes. Given that the data is not pooled, FL loses the opportunity to expand the number of rare events if the modeling is performed in siloes and only the meta-model is shared across institutions. Algorithmic bias may be harder to detect if each team only sees its own data. Given how challenging it is to detect and fix algorithmic bias in models trained on pooled data, it would likely be even more difficult to perform this crucial step when learning is distributed and de-centralized. This is not to say that FL has little role in healthcare. For certain machine learning projects, namely those that involve medical imaging, FL has demonstrated valuable contribution because of standard data formats and relative ease of the requisite data curation prior to modeling. However, modeling that involves the use of electronic health records from institutions with different information systems and with heterogeneous clinical practice patterns will pose a considerable challenge in data curation, especially if done in siloes. The collaboration established by the investigators behind the proposal, and their expertise, is best leveraged by creating a de-identified multi-institution dataset from a diverse population that is shared with the research community.

Another potential source of bias lies in the source of the data. For example, bias due to sample size for sensitive attributes such as age or race. Dataset biases in the form of prejudice, underestimation, and negative legacy, which have been studied and identified in centralized federated learning. FL has been lauded for its potential to create larger datasets of underrepresented diseases and bring algorithms to hospitals with fewer resources. [1,25] However, all the studies included in this review sourced data from institutions within developed countries. As the data for the studies were not made available, it is unknown if these datasets incorporate underrepresented cases or patients. Most FL frameworks use some form of fusion algorithms to aggregate algorithm weights from the models of partner institutions which may induce bias depending on whether the aggregation function performs an equal or weighted average. In such scenarios FL algorithms may weigh higher the contributions from populations with more data which in turn amplifies effects of over-/under-representing specific groups in a dataset. Deploying ML algorithms without a bias mitigation strategy risks perpetuating equity in healthcare delivery. [26,27]

## Reproducibility

Prior work has shown that data or code accessibility for healthcare has been limited compared to other industries. [28] Our review observed that about only half of the included studies followed the principles of reproducibility in either making their data publicly accessible and/or providing access to code or containers at time of publication. Limited access to data and/or code prevents external validation. Consequently, innovative, and potentially transformative models are less likely to be adopted. As of the writing of this review, there are several venues to facilitate external validation sharing through code repositories (e.g., 'GitHub') or operating system virtualization solutions (e.g., containers) to test models on local data. There is some irony in the observation that some of the reviewed FL implementations lack reproducibility. Going forward, it will be important for author teams to consider making metadata, code, and models available, although there are issues with making healthcare data publicly available.

## Fairness

As the quality and quantity of the data, as well as the local resources vary among the FL participating institutions, their contributions to the final FL model can also vary. This leads to undesirable risks and biases which strongly affect FL outcomes. In cases where FL incentive mechanisms deployed, this may also affect the benefits that participating institution acquire from the data federations they join. For example, the final federated model should not favor institutions that respond more expeditiously during the training process. In such situations, fairness evaluation at different stages of these FL models such as selection of participating institutions, model optimization and incentive distribution, becomes important. Fairness is a more recent research direction in FL and is still in its early stages. Some popular fairness metrics including statistical parity difference, equal opportunity odds, average odds difference and disparate impact which can be considered by investigators while adopting the FL mechanism.

## Interoperability

Interoperability is particularly relevant to FL. Inhomogeneous data distribution poses a challenge to FL efforts as similarly structured and distributed data are often assumed. Despite the use of diverse datasets and the use of parameter in the majority of the papers reviewed, only one publication explicitly identified the use of an interoperability standard, the Observational Medical Outcomes Partnership (OMOP) Common Data Model (CDM), as part of their approach. [15] There are several existing initiatives aiming to make data curation and aggregation as efficient as possible including Fast Healthcare Interoperability Resources (FHIR ®), Health Level Seven (HL7 ®) standards which serve to facilitate movement of healthcare data. FL using data from different institutions will be catalyzed by adopting such interoperability standards.

## Federated learning & privacy

While FL allows ML model building without raw data collection from multiple institutions, the possibility for an adversary to learn about or memorize sensitive user data by simply tweaking the input datasets and probing the output of the algorithm exists [29,30]. Differential privacy (DP) is a new notion that is tailored to such federated settings capable of providing a quantifiable measure of data anonymization [31]. Several techniques are being explored such as distributed stochastic gradient descent, local and meta differential privacy methods [32,33]; which essentially adds noise to preserve the user-data privacy during the federated training. While there is much focus on privacy in FL, note that there is a crucial tradeoff between the convergence of the ML models during training and privacy protection levels as better convergence comes with lower privacy. Further research on privacy-preserving FL architectures with different tradeoff requirements on convergence performance and privacy levels is therefore much desirable.

## Review limitations

There are several limitations to our study. Our search strategy includes papers that were published up through 2020. Since that time new additions to the published literature that would have otherwise been included in the review. We also concede that we were unable to compare all possible features and characteristics between papers in part owing to the complexity of the content as well as space limitations of the publication. To this end, part of our critique was on the reproducibility of the published studies. This was assessed at the time of publication, and it

is possible that the investigators published data, code and/or a repository in the period after publication.

## Conclusion

In summary, we found that federated learning has been successfully piloted using international teams spanning a variety of use cases within clinical domains relevant to computer vision and oncology as forerunners. Based on the trends in reported studies, there are opportunities to improve and build upon reproducibility and the potential for bias.

## Supporting information

**S1 Table. Preferred Reporting Items for Systematic Reviews and Meta-Analyses (PRISMA) checklist.**
(DOCX)

**S2 Table. Systematic review search strategy.**
(DOCX)

**S3 Table. Transparent Reporting of a multivariable prediction model of Individual Prognosis Or Diagnosis (TRIPOD) analysis of include studies.**
(XLSX)

**S4 Table. Prediction model Risk Of Bias Assessment Tool (PROBAST) analysis of included studies.**
(XLSX)

**S1 Fig. Prediction model Risk Of Bias ASsessment Tool (PROBAST) bias analysis.**
(TIFF)

**S2 Fig. Transparent Reporting of a multivariable prediction model of Individual Prognosis Or Diagnosis (TRIPOD) adherence analysis.**
(TIFF)

## Author Contributions

**Conceptualization:** Dana Moukheiber, Leo Anthony Celi.

**Data curation:** Matthew G. Crowson, Dana Moukheiber, Aldo Robles Arévalo, Barbara D. Lam, Sreekar Mantena, Aakanksha Rana, Leo Anthony Celi.

**Formal analysis:** Matthew G. Crowson, Dana Moukheiber, Aldo Robles Arévalo, Barbara D. Lam, Sreekar Mantena, Aakanksha Rana, Leo Anthony Celi.

**Funding acquisition:** Aakanksha Rana, David W. Bates.

**Investigation:** Matthew G. Crowson, Dana Moukheiber, Aldo Robles Arévalo, Deborah Goss, Leo Anthony Celi.

**Methodology:** Matthew G. Crowson, Dana Moukheiber, Aldo Robles Arévalo, Barbara D. Lam, Sreekar Mantena, Deborah Goss, Leo Anthony Celi.

**Project administration:** Matthew G. Crowson.

**Resources:** Matthew G. Crowson, Deborah Goss.

**Software:** Matthew G. Crowson, Deborah Goss.

**Supervision:** David W. Bates, Leo Anthony Celi.

**Validation:** Matthew G. Crowson.

**Writing – original draft:** Matthew G. Crowson, Dana Moukheiber, Aldo Robles Arévalo, Barbara D. Lam, Sreekar Mantena, Aakanksha Rana, Deborah Goss, David W. Bates, Leo Anthony Celi.

**Writing – review & editing:** Matthew G. Crowson, Dana Moukheiber, Aldo Robles Arévalo, Barbara D. Lam, Sreekar Mantena, Aakanksha Rana, Deborah Goss, David W. Bates, Leo Anthony Celi.

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
