## [Decision Letter · Decision Letter 0]

15 Mar 2022

PDIG-D-22-00011

A Systematic Review of Federated Learning Applications for Biomedical Data

PLOS Digital Health

Dear Dr. Crowson,

Thank you for submitting your manuscript to PLOS Digital Health. After careful consideration, we feel that it has merit but does not fully meet PLOS Digital Health's publication criteria as it currently stands. Therefore, we invite you to submit a revised version of the manuscript that addresses the points raised during the review process.

We look forward to receiving your revised manuscript.

Kind regards,

Dylan A Mordaunt, MB ChB, MPH, MHLM, FRACP, FAIDH

Academic Editor

PLOS Digital Health

Journal Requirements:

1. Please amend your detailed Financial Disclosure statement. This is published with the article, therefore should be completed in full sentences and contain the exact wording you wish to be published.

State what role the funders took in the study. If the funders had no role in your study, please state: “The funders had no role in study design, data collection and analysis, decision to publish, or preparation of the manuscript.”

2. Please update your Competing Interests statement. If you have no competing interests to declare, please state: “The authors have declared that no competing interests exist.”

3. Please provide a complete Data Availability Statement in the submission form, ensuring you include all necessary access information or a reason for why you are unable to make your data freely accessible. Note that it is not acceptable for the authors to be the sole named individuals responsible for ensuring data access.

PLOS defines a study's minimal data set as the underlying data used to reach the conclusions drawn in the manuscript and any additional data required to replicate the reported study findings in their entirety. Any potentially identifying patient information must be fully anonymized. 

If your research concerns only data provided within your submission, please write “All data are in the manuscript and/or supporting information files.” as your Data Availability Statement.

4. Please provide separate figure files in .tif or .eps format only and ensure that all files are under our size limit of 20MB.

For more information about how to convert your figure files please see our guidelines: https://journals.plos.org/digitalhealth/s/figures

Additional Editor Comments (if provided):

This is a great systematic review. We had difficulty identifying reviewers, but I think that is partly because systematic review and meta-analysis is more common in health sciences than math/engineering. The use of structured reporting checklists is ideal. The reviewer has provided some great comments, which I'll leave to the authors to decide how to address. I would suggest ensuring all checklists are appended and that the AMSTAR-2 is considered to check the quality of the SR (and also included).

With regards to the criteria for publication:

1) Originality - the study appears to be an original contribution.

2) High importance and broad interest to community of researchers, engineers and clinicians working in the field of digital health.

3) High methodological rigor and ethical standards.

4) Substantial evidence for its conclusions.

5) Clearly outlined utility and accessibility for the broader community.

6) Follow appropriate standards and practice of open science.

Reviewers' comments:

Reviewer's Responses to Questions

**Comments to the Author**

1. Does this manuscript meet PLOS Digital Health’s publication criteria? Is the manuscript technically sound, and do the data support the conclusions? The manuscript must describe methodologically and ethically rigorous research with conclusions that are appropriately drawn based on the data presented.

Reviewer #1: Yes

2. Has the statistical analysis been performed appropriately and rigorously?

Reviewer #1: Yes

3. Have the authors made all data underlying the findings in their manuscript fully available (please refer to the Data Availability Statement at the start of the manuscript PDF file)?

Reviewer #1: Yes

4. Is the manuscript presented in an intelligible fashion and written in standard English?

Reviewer #1: Yes

5. Review Comments to the Author

Reviewer #1: The paper "A Systematic Review of Federated Learning Applications for Biomedical Data" follows the PRISMA approach for systematic literature reviews and discusses the use of federated learning in medicine. The set of included papers is limited to actual multi-centre federated learning approaches, as opposed to the large collection of works simulating federated learning on a single device. The authors identify key challenges for the application of federated learning for healthcare, while also showing the need to further explore this research direction. I am of the opinion that the research community would benefit from the publication of this systematic review.

The paper is overall well-written with no apparent language mistakes. The structure is determined mostly by the PRISMA approach, which has been followed rigorously and correctly. All necessary items from the PRISMA checklist have been answered.

There are a few minor things that could be improved upon:

- Although the scope of the systematic review is defined in the beginning, I am missing some more remarks regarding the natural privacy (or lack thereof) of federated learning. Recent works have shown the vulnerability of FL to reconstruction attacks (e.g. https://arxiv.org/pdf/2112.02918.pdf) and other kinds of attacks, which is not mentioned at all in the review. Due to the possible leakage of sensitive patient information using FL without measures such as Differential Privacy, to me this topic is essential to mention somewhere in the review.

- The paper could also benefit from mentioning and briefly discussing existing systematic literature reviews in this field (e.g. https://www.mdpi.com/2076-3417/11/23/11191/htm), explaining the unique value of the author's review in comparison. I understand that this does not fit into the Results section of the paper, but could be included in the Discussion.

- In the section "Methods - Search Strategy & Selection Criteria", the searched literature collections are listed. To me it is not apparent why those specific libraries were selected, and not for example IEEE Xplore. If there is some reasoning behind this, it would be good to include it here.

- The table captions do not always explain the table sufficiently. For instance, in Table 2, I do not understand the meaning of the column "Number of Entries" compared to the column "Patient Cohort Size". Please clarify in the table caption or rename the column appropriately.

- In Figure 1 for the number entry for "Records after duplicates removed", the authors used the PRISMA flowchart slightly incorrectly. I believe the number n corresponds to the number of papers still included after duplicate removal, which would be 2171, as in the box for "Records screened".

- In Table 2 for the "Number of Entries" for "Huang et al.", the thousands separator is shifted one digit to the left.

If those items are addressed in the paper or there are very good reasons why they have been excluded, I am happy to accept this paper for publication.

6. PLOS authors have the option to publish the peer review history of their article (what does this mean?). If published, this will include your full peer review and any attached files.

**Do you want your identity to be public for this peer review?** For information about this choice, including consent withdrawal, please see our Privacy Policy.

Reviewer #1: No

---

## [Editor Report · Decision Letter 1]

30 Mar 2022

A Systematic Review of Federated Learning Applications for Biomedical Data

PDIG-D-22-00011R1

Dear Dr. Crowson,

We are pleased to inform you that your manuscript 'A Systematic Review of Federated Learning Applications for Biomedical Data' has been provisionally accepted for publication in PLOS Digital Health.

Best regards,

Dylan A Mordaunt

Academic Editor

PLOS Digital Health

Thank you for your resubmission. This now meets the criteria for publication. I think this is a valuable addition to the field.